# Investigating the effect of relationship satisfaction on postpartum women's health-related quality of life in Burkina Faso: a cross-sectional analysis

Paul Lokubal [ID],[1,2] Clara Calvert [ID],[2] Simon Cousens [ID],[3] Marina Daniele [ID],[4] Rasmané Ganaba,[5] Veronique Filippi [ID][3]

[1]Department of Population Health, University of Oxford, Oxford, UK
[2]Department of Population Health, London School of Hygiene and Tropical Medicine, London, UK
[3]Department of Infectious Disease Epidemiology, London School of Hygiene and Tropical Medicine, London, UK
[4]Women and Children's Health, King's College London, London, UK
[5]Agence de Formation de Recherche et d'Expertise en Santé pour l'Afrique (AFRICSanté), Bobo Dioulasso, Burkina Faso

**Correspondence to**
Dr Paul Lokubal;
paul.lokubal@ndph.ox.ac.uk

## ABSTRACT

**Introduction** The period following childbirth poses physiological, physical, social and psychological challenges to women that may affect their quality of life. Few studies in Africa have explored women's health-related quality of life (HrQoL) and its determinants in postpartum populations, including the quality of women's relationships with their male partners. We investigated whether relationship satisfaction was associated with better HrQoL among postpartum women in Burkina Faso, 8 months after childbirth.

**Methods** We analysed data from 547 women from the control arm of a randomised controlled trial in Burkina Faso. The study outcome was a woman's HrQoL, assessed using the cross-culturally validated WHOQOL-BREF tool, with response categories adapted for Burkina Faso. The exposure was relationship satisfaction measured using questions adapted from the Dyadic Adjustment Scale and Marital Assessment Test tools. We calculated the median HrQOL scores for the study sample, overall and for each domain of HrQOL (physical, psychological, social and environmental). The association between relationship satisfaction and HrQoL was examined using multiple linear regression models with robust SEs.

**Results** Postpartum women had high median HrQoL scores in the physical (88.1), psychological (93.1), social (86.1) and environmental (74.0) domains and overall HrQoL (84.0). We found that higher relationship satisfaction is associated with increased HrQoL. After adjusting for potential confounders, we found that for each point increase in relationship satisfaction score, the increase in HrQoL was 0.39 (p<0.001) for the overall HrQoL; 0.32 (p=0.013) for the physical domain; 0.25 (p=0.037) for the psychological domain; 0.46 (p<0.001) for the social domain and 0.49 (p<0.001) for the environmental domain.

**Conclusion** Higher relationship satisfaction is associated with higher HrQoL scores. Policies should aim to support women to cope with the challenges of childbirth and childcare in the postpartum period to improve postpartum women's HrQoL.

## Strengths and limitations of this study

► This is one of the few studies to explore health-related quality of life (HrQoL) during the extended postpartum period in an African setting, whereas most studies have documented it during pregnancy or the immediate postpartum period.

► Data collection on HrQoL and relationship satisfaction took place at one point in time 8 months after childbirth and we cannot rule out reverse causality in which women who have high HrQoL scores are consequently more likely to feel satisfied with their relationship.

► We did not validate the tools locally for collecting the outcomes and the main exposure. However, we adapted them to suit the local context and also conducted rigorous pretesting to ensure it was adequately understood by the participants.

well-being and not merely the absence of disease or infirmity',[1] revolutionised the concept of health and led to major advances in its assessment beyond traditional health status indicators of morbidity and mortality. This definition led to the development of subjective concepts such as quality of life (QoL).[2] According to WHO, QoL is defined as 'individuals' perceptions of their position in life in the context of the culture and value systems in which they live and in relation to their goals, expectations, standards and concerns'.[2 3] It is a multidimensional concept that captures all spheres of a person's life; it includes physical, mental, social and functional aspects of an individual's life and their satisfaction with it.[4] According to Torrance, health-related quality of life (HrQoL) is 'a subset [of QoL] relating only to the health domain of that [person's] existence'.[5]

Following childbirth and during the postpartum period, many women experience physical, emotional, functional, social and

## INTRODUCTION

The 1948 definition of health by the World Health Organisation (WHO) as 'a state of complete physical, mental and social

psychological challenges that can affect their well-being.[6–10] For example, one study of women 9–12 months post partum in the USA found that 69% of women experienced at least one physical symptom following childbirth,[7] while another study conducted in Spain reported that 85% of women experienced fatigue 6 weeks after childbirth.[8] Some of these challenges are known to persist well beyond the traditional 6-week postpartum period and are associated with poor overall HrQoL among postpartum women.[7 11]

The reasons for poor HrQoL noted among women in the postpartum period are complex and are likely to be driven by a range of different factors including, for example, clinical experiences (ie, if any complications occurred during delivery), and sociodemographic and economic characteristics.[11–16] There is a growing body of evidence pointing to the important role that relationship quality plays in HrQoL for women, including during pregnancy and in the postpartum period.[16–20] The relationship between relationship satisfaction and QoL is complex and maybe reciprocal in which relationship satisfaction may influence or be influenced by QoL.[21 22] There are many possible mechanisms through which higher relationship quality may lead to improved HrQoL among postpartum women including existence of emotional connection between spouses which helps to protect against postpartum depression[16 23]; easier resolution of conflicts and sharing of childcare responsibilities,[12] and social, financial and emotional support from the male partner following childbirth and during the postnatal period.[24 25] There are only a few, generally small, studies that have directly examined the influence of relationship satisfaction on women's postpartum HrQoL. In these studies, spousal support,[24 25] staying with the partner post partum[26] and women's marital satisfaction with their spouses,[16–20] are associated with higher HrQoL among postpartum women.

Improving maternal well-being is a key priority with the Sustainable Development Goals agenda.[27] Given the paucity of studies conducted in sub-Saharan Africa on women's postpartum HrQoL and the interest in the impact of male partner involvement on women's health and well-being,[28] we analysed data from 547 women who were enrolled in the control arm of a randomised controlled trial (RCT) to investigate the effects of relationship satisfaction on postpartum women's HrQoL in Burkina Faso.

## METHODS
### Dataset and study design
The study is a secondary analysis of data from the control arm of an individual, non-blinded RCT conducted in five primary healthcare facilities in three health districts in the city of Bobo-Dioulasso, Burkina Faso. The data were collected between 16 February 2015 and 4 July 2016. The details of the study design, data collection methods and data management, and ethical approval are described

in detail elsewhere.[28] In brief, the trial investigated the effectiveness of a health services intervention designed to increase male partners' involvement in maternity care, hypothesising that the intervention would increase postnatal care attendance, the duration of exclusive breast feeding and the use of postpartum contraception.[28] Women were eligible for the trial if they were aged 15–45 years and living with a male partner—either cohabiting or legally married.[28] The RCT intervention was provided only to named individuals who received personalised invitations to attend a series of three educational sessions in the health facility, and health workers checked the names of those attending against a list of people invited to minimise contamination between the study arms due to social interactions between individuals in the community. A total of 1114 women (583 in the intervention arm and 561 in the control group) were enrolled in the study and followed from around 20 weeks of gestation up to 8 months post partum.[28]

### Patient and public involvement
This was a secondary data analysis of a primary study and while full participant consent was obtained in the primary study, it was not required for this analysis and publication. The dataset has been published, as required by the funder (DFID) on the London School of Hygiene and Tropical Medicine Data Compass repository following full de-identification (eg, all personal names removed, dates changed to month/year format and health centre names anonymised). Researchers can apply to use the data which can be released at the discretion of the study authors. Only the variables needed for the proposed secondary analyses are released.

### Variables
#### Outcome: HrQoL
The outcome for this analysis is HrQoL and its four domains, measured at 8 months post partum.[3] Data on HrQoL were collected using the French version of the WHO Quality of Life-BREF (WHOQOL-BREF) tool,[2] which has been validated among French-speaking populations.[29] This tool has been validated for use in the general population in Africa, but not specifically Burkina Faso, so response options were adapted to the local context by the study authors to reflect the low formal education level among women and to address difficulties in differentiating between response options in local languages.[28] All the questions were translated to and back translated from French into Dioula and Moore during training workshops involving the principal investigator, the local field supervisor and the five interviewers. The questionnaires were then pretested by the interviewers with acquaintances or women in their neighbourhood. They were asked to report back on the intelligibility and cultural appropriateness of the proposed question formulation. Following their feedback, small further modifications were made.

The WHOQOL-BREF is a 26-item QoL assessment tool covering four domains: physical (seven items—enquired

about pain and discomfort; energy and fatigue; sleep and rest; dependence on medication; mobility; activities of daily living and working capacity), psychological (six items—enquired about positive feelings; negative feelings; self-esteem; thinking learning; memory and concentration; body image; spirituality, religion and personal beliefs), social (three items—personal relations; sex and practical social support) and environmental (eight items—financial resources; information and skills; recreation and leisure; home environment; access to health and social care; physical safety and security; physical environment; and transport). There are two further questions on general QoL and general health.[4] The items probed women's perceptions about their life in the 4 weeks preceding the questionnaire. For this study, each item was rated on a 3-point Likert scale (ie, dissatisfied/low=1, neither satisfied nor dissatisfied/moderate/average=2, satisfied/very much/high=3) with lower scores indicating lower HrQoL and the higher scores higher HrQoL. We summed these scores to give an overall HrQoL score and individual domain scores for each individual. We then transformed the respective scores to a 20-point scale and subsequently to a 100-point score scale as per WHO instructions,[3] and analysed it as a continuous variable.

### Exposure: relationship satisfaction

The primary predictor for this study was relationship satisfaction measured at 8 months post partum. We defined 'relationship satisfaction' in the postpartum period as a 'woman's satisfaction with her relationship with her partner and the degree of communication, shared decision-making and agreement between the couple'.[28] Data for this variable was only collected for women who were still staying with their partners or husbands 8 months after childbirth. If women became single by 8 months post partum, data on relationship satisfaction were not collected and therefore these women are not part of the sample analysed. We calculated the relationship satisfaction score using a combination of 35 pretested questions adapted from the Marital Adjustment Test and Dyadic Adjustment Scale,[30 31] but not locally validated for use in Burkina Faso. Women were asked about their satisfaction with their partner, satisfaction with degree of communication, satisfaction with shared decision-making and agreement on 12 variables (family finances, the woman's own family, the woman's in-laws, number of children, children's health, children's education, children's nutrition, man's work, woman's work, time the couple spent together, family planning and cospouses or other women). Lower scores on these questions indicate lower satisfaction. We summed the scores from each individual question across all the questions to generate the total score. We analysed relationship satisfaction as a continuous variable.

### Potential confounders

We considered the following variables as potential confounders of the association between relationship satisfaction and HrQoL: women's age at enrolment (15–24, 25–34 and 35+); women's education level (none, primary and secondary or higher); male partner's age at enrolment (20–29, 30–39 and 40+); male partner's education level (none, primary, secondary and tertiary); type of relationship (polygamy and monogamy); women's parity (primiparous—first child for that childbirth, secundipara—second child for that childbirth, multipara—more than two children for that childbirth); woman's postpartum place of residence (with her parents/in-laws and with male partner); source of finance for women's healthcare after childbirth (divided into four categories of did not attended care, paid for herself, care was paid for by partner and care was paid for by others); woman's need for contraception (not met and met); and woman's views on domestic violence (justified for at least one reason and not justified at all).

### Statistical analysis

We examined the distribution of participants' characteristics at baseline and at 8 months using percentages for categorical variables and means, SD, medians and IQRs for continuous variables. We excluded missing data as it was minimal.

We examined the internal consistency and reliability of both the WHOQOL-BREF tool and relationship satisfaction index using Cronbach's alpha coefficient, which assesses how well a set of items, variables or attributes measures a single, one-dimensional latent construct among participants involved in a study. We considered Cronbach's alpha coefficient values 0.70 and above satisfactory.[32] We then calculated the median and IQRs for the HrQoL (overall score and for each domain). This was done for the total sample of women and also stratified by relationship satisfaction. We conducted Spearman's rank correlation to investigate any correlation between the domain scores and relationship satisfaction.

In the multivariate analysis, we considered all the other variables potential confounders for the association between relationship satisfaction and HrQoL. Due to presence of heteroskedasticity, we ran a multiple linear regression model with robust SEs to estimate adjusted regression coefficients, adding all the variables to the model simultaneously. We analysed the data using STATA/SE V.15.1.

### RESULTS

Of the 561 women enrolled in the control arm of the study, 14 were missing at the 8 months' post partum follow-up period: 11 were lost-to-follow-up without further information, one moved away, one withdrew voluntarily from the study and one died. Thus, the analysis is based on 547 women who were still enrolled in the study at 8 months post partum.

Table 1 shows women's and their partner's characteristics at baseline and at 8 months. The relationship satisfaction score ranged from 3 to 56, with a mean score of 30.5

**Table 1** Demographic characteristics of women at the baseline and at 8 months (number, N=547)

| Variable | Number (n) | Percent | Mean | SD |
|---|---|---|---|---|
| Relationship satisfaction* | 533 | | 30.5 | 10.1 |
| Age (in years) | | | | |
| 15–24 | 233 | 42.6 | | |
| 25–34 | 250 | 45.7 | | |
| 35+ | 64 | 11.7 | | |
| Education level reached | | | | |
| None | 271 | 49.5 | | |
| Primary | 163 | 29.8 | | |
| Secondary or higher | 113 | 20.7 | | |
| Type of relationship | | | | |
| Polygamous | 84 | 15.4 | | |
| Monogamous | 463 | 84.6 | | |
| Woman's parity | | | | |
| Primiparous | 139 | 25.4 | | |
| Secundipara | 222 | 40.6 | | |
| Multiparous | 186 | 34.0 | | |
| Male partner's age (in years)† | | | | |
| 20–29 | 133 | 27.0 | | |
| 30–39 | 240 | 48.8 | | |
| 40+ | 119 | 24.2 | | |
| Male partner's education level reached‡ | | | | |
| None | 236 | 47.9 | | |
| Primary | 122 | 24.8 | | |
| Secondary | 87 | 17.7 | | |
| Tertiary | 48 | 9.7 | | |
| Woman's postpartum place of residence§ | | | | |
| In-laws/own family | 82 | 15.2 | | |
| Husband | 457 | 84.8 | | |
| Funding for woman's care§ | | | | |
| Did not attend care | 91 | 16.9 | | |
| Woman | 63 | 11.7 | | |
| Partner | 346 | 64.2 | | |
| Others | 39 | 7.2 | | |
| Meet need for family planning§ | | | | |
| Unmet | 101 | 18.7 | | |
| Met | 438 | 81.3 | | |
| Views on domestic violence§ | | | | |
| Justified for at least one reason | 210 | 39.0 | | |

Continued

**Table 1** Continued

| Variable | Number (n) | Percent | Mean | SD |
|---|---|---|---|---|
| Not justified at all | 329 | 61.0 | | |

*Data on only 533 observations available.
†Data on only 492 observations available.
‡Data on only 493 observations available.
§Data on only 539 observations available.

(SD=10.1), median score of 30 and IQR of 13 (24–37). The Cronbach's alpha coefficient for relationship satisfaction was 0.849 (not shown). The youngest woman was 16 years and the oldest was 43 years old. The median age of the women was 26 years with an IQR of 9 years (21–30 years) and only 64 (12%) were 35 years and older. About 50% of women had no formal education and 85% were in a monogamous relationship. There was information available on 492 male partners with a median age of 33 years (IQR 29–39 years). The youngest was 20 years and the oldest was 62 years old and about half (49%) of the men were aged 30–39 years. Over 70% (358) of the men had either no education or had attained primary education only. Eighty-five per cent (457) of women stayed at the partner's place post partum, 64% (346) had their healthcare paid for by their partners while 81% (438) had met the need for family planning while 61% did not view domestic violence justifiable at all.

Table 2 shows transformed HrQoL scores with Cronbach's alpha coefficient (N=547). Overall, we observed excellent internal consistency across all 26 questions with a Cronbach's alpha coefficient of 0.853. For the individual domains, Cronbach's alpha was as follows: 0.703 for physical, 0.721 for psychological, 0.383 for social and 0.555 for environmental. In this population, the median overall HrQoL score was 84.0 (IQR 14.4).

Table 3 shows the Spearman's rank correlation results between the domains of HrQoL. We noted statistically significant moderate correlation between the physical and psychological domains (58%, p value <0.001), physical and environmental domains (56%, p<0.001), psychological and environmental domains (58%, p value <0.001), psychological and social domains (47%, p value <0.001), and social and environmental domains (50%, p value <0.001).

Table 4 shows the unadjusted and adjusted mean regression coefficients with robust SEs for the association between HrQoL and relationship satisfaction. In both unadjusted and fully adjusted models, relationship adjustment was associated with the overall HrQoL and its domains. We found that for each point increase in relationship score, the overall HrQoL score increased by 0.39 (p value <0.001) in both the unadjusted (95% CI 0.23 to 0.56) and adjusted regression models (95% CI 0.20 to 0.57). In the unadjusted model, the highest increases in HrQoL were noted with the social (regression

**Table 2** Transformed health-related quality of life scores with Cronbach's alpha coefficient (N=547)

| Domains | Number | Mean | SD | Median | 25th percentile | 75th percentile | Cronbach's alpha |
|---|---|---|---|---|---|---|---|
| Physical* | 546 | 85.0 | 12.9 | 88.1 | 82.1 | 94.0 | 0.703 |
| Psychological† | 545 | 88.4 | 13.4 | 93.1 | 79.2 | 100 | 0.721 |
| Social† | 545 | 88.0 | 14.0 | 86.1 | 86.1 | 100 | 0.383 |
| Environmental† | 545 | 74.5 | 12.9 | 74.0 | 63.5 | 84.4 | 0.550 |
| Overall‡ | 540 | 83.0 | 10.5 | 84.0 | 76.0 | 90.4 | 0.853 |

*One observation missing.
†Two observations missing.
‡Seven observations missing.

coefficient=0.56, 95% CI 0.35 to 0.77, p value <0.001) and the environmental domains (regression coefficient=0.50, 95% CI 0.30 to 0.69, p value <0.001) while the lowest increases were noted with the psychological (regression coefficient=0.25, 95% CI 0.05 to 0.44, p value =0.013 and physical (regression coefficient=0.28, 95% CI 0.07 to 0.49, p=0.010). There were no differences in regression coefficients between the adjusted and unadjusted models in the psychological and the environmental domains while the regression coefficient for the physical domain increased to 0.32 (95% CI 0.07 to 0.57, p value =0.013) but that for the social domain decreased to 0.46 (95% CI 0.22 to 0.69, p value <0.001).

## DISCUSSION
### Key findings
Overall, this study shows that higher relationship satisfaction scores are associated with better HrQoL among postpartum women in Burkina Faso for all the individual domains of HrQoL and the overall HrQoL.

The dose–response association between relationship satisfaction and the overall HrQoL and its domains is consistent with findings from studies elsewhere that demonstrate that higher marital or relationship satisfaction is associated with an improved HrQoL among postpartum women.[11 17 33–36] These findings are also consistent with findings from a systematic review exploring what matters to women in the postnatal period that showed that family relationships as well as overcoming physical and emotional challenges in the postnatal period were important considerations for women.[37] Higher relationship satisfaction is likely to be related to women's assessments of better spousal social, economic and emotional

support in the postpartum period, factors known to be associated with higher HrQoL overall.[24 26 38–40] Better spousal support has been shown to lead to reductions in stress, anxiety during pregnancy, childbirth and in the postpartum period, offsetting some of the challenges of coping with parenthood and child care.[26] In our study, we defined relationship satisfaction in relation to shared decision making, agreement on key family issues and communication between the couples. Couples who frequently discuss, agree and make joint decisions on key family issues are likely to support each other to achieve their desired targets and thus have a higher HrQoL than those who do not.[24 25] Increased spousal support may be responsible for the observed association between better relationship scores with HrQoL. However, women in Burkina Faso typically receive support from their extended family and social network, often in the form of financial and other assistance (such as helping in the care of the newborn) following childbirth.[9] Thus, factors other than spousal support may explain the high HrQoL reported by women in our study. It is also plausible that a positive feedback loop or reciprocal relationship exists between relationship satisfaction and HrQoL in which relationship satisfaction influences and is influenced by HrQoL,[21 22] in which, for example, better relationship satisfaction leads to better HrQoL which in turn leads to better relationship satisfaction.

Our results also show that overall, women had high HrQoL scores at 8 months post partum. In a study validating the WHOQOL-BREF tool at 6 weeks post partum among Australian women with postnatal depression (PND) and those without PND, women without PND had higher mean scores compared with those with PND; the

**Table 3** Correlation matrix between health-related quality of life domains

| | Physical | Psychological | Social | Environmental |
|---|---|---|---|---|
| Physical | 1.00 | | | |
| Psychological | 0.58 (<0.001) | 1.00 | | |
| Social | 0.36 (<0.001) | 0.47 (<0.001) | 1.00 | |
| Environmental | 0.56 (<0.001) | 0.58 (<0.001) | 0.50 (<0.001) | 1.00 |

P values in parentheses.

**Table 4** Association between relationship satisfaction and HrQoL using multiple linear regression with robust SEs*

| Relationship satisfaction | Unadjusted regression coefficients | | | Adjusted regression coefficients* | | |
|---|---|---|---|---|---|---|
| | Coeff. | 95% CI | P value | Coeff. | 95% CI | P value |
| Physical | 0.28 | 0.07 to 0.49 | 0.010 | 0.32 | 0.07 to 0.57 | 0.013 |
| Psychological | 0.25 | 0.05 to 0.44 | 0.013 | 0.25 | 0.02 to 0.49 | 0.037 |
| Social | 0.56 | 0.35 to 0.77 | <0.001 | 0.46 | 0.22 to 0.69 | <0.001 |
| Environment | 0.50 | 0.30 to 0.69 | <0.001 | 0.49 | 0.25 to 0.73 | <0.001 |
| Overall HrQoL | 0.39 | 0.23 to 0.56 | <0.001 | 0.39 | 0.20 to 0.57 | <0.001 |

*Adjusted for age, education level, type of relationship, parity, male partner's age, male partner's education level, woman's postpartum place of residence, funding for woman's healthcare, met need for contraception and views on domestic violence.
Coeff., regression coefficient; HrQoL, health-related quality of life.

mean overall score for HrQoL for women without PND was 15.8 in the 20-score scale (equivalent to 73.9 in the 100-score scale) and the highest mean score of HrQoL for women without PND was observed in the environmental domain (16.2 in the 20-score scale, equivalent to 76.3 when transformed to the 100-score scale) while the lowest was observed in the psychological scale (15.5/20 or 71.9/100).[41] In our study, however, the highest mean score was observed in the psychological domain with the lowest obtained in the environmental domain. The differences are likely due to sociodemographic and economic differences as well as differences in infrastructure, accessibility of local health-promoting resources and amenities of the local environments between Australian women (high-income country) and those from our study (low-income country). Whereas most studies show that women's postpartum HrQoL decreases following childbirth, this period also often brings joy to women and their partners and may be associated with improved HrQoL.[42] It is also possible that any childbirth related-health issues, for example, postpartum depression, will have resolved by 8 months and women may have adapted to the demands of caring for a new infant. Other studies suggest that HrQoL is related to time from childbirth. A longitudinal study comparing perinatal and postnatal HrQoL among 465 women in semi-rural Bangladesh showed that overall mean HrQoL scores increased by 37% from the perinatal period (within 2–7 days) to the postpartum period (within 40–60 days).[43] The same study also showed that whereas 58% of women experienced moderate or severe levels of health problems in the perinatal period, these (health problems) resolved in the postpartum period and women experienced a high QoL.[43] Another prospective longitudinal study of Iranian women at 6–8 and 12–14 weeks post partum showed that women had higher HrQoL scores in the 12–14 weeks' period compared with 6–8 weeks.[44] In contrast, a study comparing HrQoL among employed and unemployed Nigerian women at 6, 12 and 18 weeks post partum showed that all women had the highest HrQoL scores at 12 weeks irrespective of employment status while employed women—with only 3 months' maternity leave—scored lower than the unemployed women at all three timepoints, underscoring the importance of other

determinants such as women's employment in addition to temporal influences on postpartum women's HrQoL.[45] However, the duration from childbirth to the time of HrQoL assessment differs markedly between women in our study (8 months) from most of the women studied.

The Cronbach's alpha coefficient was satisfactory for the overall HrQoL, and the physical and psychological domains, but low for the social and environmental domains. This may be related to the small numbers of items for the social domain, the unrelatedness of the questions to a particular domain among this population or the heterogeneity of the questions.[2 46 47] Although the internal consistency of the WHOQOL-BREF tool has been established in the general population,[2] and among postpartum women,[41] and found to be satisfactory, it has not been validated among postpartum women in Burkina Faso or Africa. Using a 3-Likert WHOQOL-BREF questionnaire on a large sample of 1963 Nigerian adolescents, Akpa and Fowobaje showed that a two-factor model of the WHOQOL-BREF in Nigeria and related settings had better internal consistency and reliability than the current universally accepted and used four-factor model.[47] It is thus possible that the differential conceptualisation of the WHOQOL-BREF tool questions,[47] may account for the low Cronbach's alpha in the social and environmental domains in our study population.

### Study strengths and limitations

This is one of the few studies to explore HrQoL during the extended postpartum period in an African setting, whereas most studies have documented it during pregnancy or the immediate postpartum period.

Our study has several limitations. It was conducted in Bobo-Dioulasso, an urban setting where access to healthcare is relatively easy and does not therefore represent the experience of women in the entire country. Data collection on HrQoL and relationship satisfaction took place at one point in time 8 months after childbirth and we cannot rule out reverse causality in which women who have high HrQoL scores are consequently more likely to feel satisfied with their relationship. In addition, we did not validate the tools locally for collecting the outcomes and the main exposure although we adapted them to suit

the local context and also conducted rigorous pretesting to ensure it was adequately understood by the participants. However, both tools had good internal reliability consistency (relationship satisfaction 0.849 and HrQoL 0.853) while the results for the social domain need to be interpreted with caution due to its low Cronbach's alpha. Although we did not compare the 5-rated and 3-rated Likert scales, the good internal consistency of the 3-rated Likert scale suggests that it may be preferable in situations where many variables need to be collected in a short time where resources are limited and participants have low education levels, which may present semantic challenges that make it difficult to distinguish between some response options in the questionnaire. However, a validation of the WHOQOL-BREF tool for use among postpartum women in sub-Saharan Africa is needed.

Our study may have also suffered from several response biases such as social desirability bias, satisficing, acquiescence and positive biases. Finally, we did not include in the analysis all potential confounders of the association between relationship satisfaction and HrQoL such as sex of the babies, delivery mode and employment that are known to affect HrQoL among postpartum women. This was due to the small number of observations in some categories, the long-time lag between the occurrence of an event (such as type of delivery) and the administration of the questions, and most of our study participants did not have meaningful gainful employment and as such, categorisation was difficult.

## CONCLUSION

Relationship satisfaction is associated with improved HrQoL among postpartum women in Burkina Faso, an important finding for public health practitioners who seek to improve women's well-being after childbirth. In another study, we found that an intervention promoting male partner involvement in maternity care in the same population improved relationship satisfaction, alongside uptake of maternity care. It is therefore plausible that policies that support male involvement in maternal health might also lead to increased HrQoL directly or through relationship satisfaction. Further observational and intervention research would be needed to confirm this link and other determinants of women's HrQoL.

**Acknowledgements** The authors wish to acknowledge the Economic and Social Research Council (ESRC) which funded Marina Daniele's studies at the London School of Hygiene and Tropical Medicine and the Chevening Scholarship and the London School of Hygiene and Tropical Medicine which funded Paul Lokubal's MSc studies at the London School of Hygiene and Tropical Medicine.

**Contributors** MD and VF conceived the idea, planned and designed the primary study. PL wrote the first draft while SC reviewed and provided statistical input. CC, SC, MD, RG and VF edited the draft and provided critical insights. All authors have approved and contributed to the final submitted manuscript.

**Funding** This study was funded by the Strengthening Evidence for Programming on Unintended Pregnancy (STEP UP) Research Programme Consortium (grant code EPIDHC20) of which VF is a member, and by a contribution from the Faculty of Epidemiology and Population Health, London School of Hygiene and Tropical Medicine. STEP UP is funded by UKAid from the Department for International Development.

**Competing interests** None declared.

**Patient consent for publication** Not required.

**Ethics approval** The original study obtained ethics approvals from the London School of Hygiene and Tropical Medicine (LSHTM) (Reference Number 8787), Population Council (Reference Number 662) and Burkina Faso Comite D'Ethique Pour La Recherche En Sante (Reference Number Decembre 2014). This was a secondary data analysis of a clinical trial registered with ClinicalTrials.gov. The ethics for this project was obtained from LSHTM MSc Ethics and Research Committee (Reference Number 16964).

**Provenance and peer review** Not commissioned; externally peer reviewed.

**Data availability statement** All data relevant to the study are included in the article. All data relevant to the analysis are uploaded as supplementary information and is with the editors. Re-use is permitted given the authors are acknowledged in any publication. Further data may be obtained by emailing MD.

**ORCID iDs**
Paul Lokubal http://orcid.org/0000-0002-1212-2035
Clara Calvert http://orcid.org/0000-0003-3272-1040
Simon Cousens http://orcid.org/0000-0001-8970-2305
Marina Daniele http://orcid.org/0000-0002-5666-9489
Veronique Filippi http://orcid.org/0000-0003-1331-3391

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
