## [Reviewer comments · BMJ Open]

ARTICLE DETAILS

TITLE (PROVISIONAL)	Investigating the effect of relationship satisfaction on postpartum women's health-related quality of life in Burkina Faso: a cross-sectional analysis
AUTHORS	Lokubal, Paul; Calvert, Clara; Cousens, Simon; Daniele, Marina; Ganaba, Rasmané; Filippi, Veronique

VERSION 1 – REVIEW

REVIEWER	Tissera, Hasagani McGill University
REVIEW RETURNED	21-Mar-2021

GENERAL COMMENTS	The authors used a unique longitudinal dataset to examine the effects of relationship satisfaction on women's postpartum health-related quality of life. The study employed a large sample size to test this question. The manuscript was clear and well-written. I do have some comments and recommendations, which are outlined below. 1) In the introduction, the authors mention that there is already research examining the link between relationship satisfaction and HrQoL. Therefore, the authors could should make a more compelling case as to why the present research is novel, and what the contributions to the field are.2) In terms of the analyses, can the authors provide a justification as to why they summed the relationship satisfaction score and created categories for analyses instead of using the continuous variables?3) The authors employed multiple regression analyses for their main analyses. I wondered whether the different domains of the HrQoL index were highly correlated, which could lead to biased results. Perhaps the authors could include a correlation matrix in which they report the associations between domains, the links between the domains and the other confounders that were included in the analyses.4) Relatedly, having relationship satisfaction in this correlation matrix could be beneficial to see how it relates to each of the domains on its own (without controlling for the confounders). I also wondered whether the link between relationship satisfaction and the psychological well-being domain was significant in univariate
--

	analyses and whether it is the parsing out of the shared variance between this and the other variables in the analyses that weakened this association. 5) Do the authors have a variable that is indicative of whether the women were primiparous or not? This would be an important confounder to control for. If the authors do not have this variable in the data, they could acknowledge this in the discussion section and speculate whether the present results would generalize to both primiparous and multiparous women. 6) Have the authors considered whether the type of marriage (polygamous vs monogamous) could moderate the link between relationship satisfaction and HrQoL? 7) Can the authors discuss and interpret the effect sizes that were found in the present study and compare it to past work? Are these effect sizes comparable to previous work? If not, can the authors speculate as to why? 8) In the abstract and the conclusion section, the authors recommend implementing more policies that are aimed at increasing male involvement during the process of childbirth and child care during the postpartum period. However, in the present study, only women's perception was measured and therefore, it is not clear whether the findings are driven by actual male involvement. Perhaps, simply perceiving that the partner is involved could be sufficient to produce the results. The authors could discuss this possibility in greater detail in the discussion section, and what it could mean in terms of the type of policies that should be encouraged.
--	---

REVIEWER	Comrie-Thomson, Elizabeth Burnet Institute
REVIEW RETURNED	03-May-2021

GENERAL COMMENTS	Recommendation: Minor revision Overarching comments The manuscript clearly presents a straightforward analysis that contributes to a more sophisticated understanding of the relationship between women's satisfaction with their coparent relationship and their overall wellbeing. This is an important contribution to the broader literature on impacts of interventions to influence male partners' engagement in maternal and newborn health, and the authors may wish to emphasise this in the Conclusion. There is some additional explanation and clarification to be added, mainly relating to statistical methods and the construction of variables. In particular, I suggest that the authors provide further justification for their choices in constructing the exposure variable. Major revisions
--

None. The authors are to be congratulated on their work.

Minor essential revisions

1. Page 6, lines 10-38: While I appreciate that the study design is described fully elsewhere, for the analysis in this paper I think it would be appropriate to clarify whether (and on what basis) the authors are confident that there was no contamination between intervention and control arms. And was the analysis an intention-to-treat or per protocol analysis?

2. Page 7, lines 6-9: Can the authors please provide more detail on why informed consent was not required for participant data to be used in this analysis and publication? For example, did participants' informed consent for the original study include consent for their de-identified data to be used for secondary analyses?

3. Page 7, line 25: Has the French version of WHOQUOL-BREF been validated in French-speaking populations? I assume this is the case but it should please be clarified.

4. Page 8, line 32: It is not clear whether the adapted items were pretested in the population where they were to be used. Could this be specified? And were items revised following pretesting? How were translation and back-translation managed during the pre-testing and revision process?

5. Page 8, line 41: Please clarify whether "agreement" refers to women's report of the extent to which she and her male partner agree on the listed topics

6. Page 9, line 2: The division of categories is not clear to me. Is it based on quartiles of scores, or total score?

7. Methods: I wonder why relationship satisfaction was not used as a continuous variable – can the authors explain their choice to construct relationship satisfaction as a categorical variable based on externally-derived categories? Because relationship satisfaction is a novel composite variable based on multiple items that have not been locally validated (and I also think the authors have not explored the internal structure of the composite variable, e.g. through item response theory?) it seems to me that measuring relationship satisfaction as a continuous variable might be preferable to dividing it according to externally-derived categories.

8. Page 9, line 23: I gather that women's postpartum place of residence was a binary variable: residing with her parents/in-laws, or residing with her husband. Either here or in the Introduction, could the authors please explain why it's meaningful to combine residing with her own parents and residing with her in-laws? And can the authors please clarify whether the husband would also reside with the woman while she is at her parents/in-laws – was this data collected?

9. Were partnered women who were not married, or single women, eligible to participate in the RCT, and in the secondary analysis? What happened to participants if a woman's relationship or marital status changed during the study period? Can the authors report any data on number of women excluded from the study due to relationship or marital status? This should also be considered in the interpretation of findings. It is surprising to me that all women in the study were married (according to Table 1).

10. Page 12, lines 32-41: Since Cronbach's alpha is designed to report on internal consistency for a single latent concept, then my understanding is that the alpha for the overall HrQoL is less meaningful than the domain-specific alpha coefficients. This should be clarified in the Results section. I also suggest that internal consistency for the overall HrQoL not be described as "excellent" without also providing a value judgement for the domain-specific alpha coefficients. I also think that the implications of the low alpha coefficients for social and environmental HrQoL domains should be addressed in the Discussion section.

11. Methods: The authors report on linear associations between relationship satisfaction and HrQoL. Were non-linear relationships tested? Were various associations explored (e.g. dot plot reviewed visually) to confirm whether a linear association was the most appropriate relationship to test?

12. Page 15, lines 12-16: Can the authors include one or more citations to support this interpretation of the mechanisms linking shared decision-making, couple communication, and agreement with HrQoL?

13. Page 15, lines 31-54 (and continuing on next page): I recommend that the authors consider whether positive feedback loops between higher psychological HrQoL and improved relationship satisfaction (and/or negative feedback loops between lower psychological HrQoL and reduced relationship satisfaction) may provide a plausible explanation for the weaker observed linear association between the psychological domain of HrQoL and relationship satisfaction. The implication of this is that psychological wellbeing, and related concepts of emotional intimacy and emotional support, may be a particularly important aspect of relationship satisfaction among the study population (consistent with the majority of the literature on relationship satisfaction and HrQoL as noted by the authors), rather than a relatively unimportant aspect of relationship satisfaction as currently hypothesised by the authors. I appreciate that the authors note as a study limitation the fact that they cannot rule out reverse causality (page 18, lines 2-4), however I think that this deserves further interrogation in the Discussion section – particularly when seeking to interpret the weaker association seen for the psychological HrQoL domain.

14. Page 16, line 17 onwards: When comparing study results with findings from other studies, could the authors please describe the timepoint at which other studies' results were captured – i.e. how many months postpartum?

Discretionary, but recommended, revisions

15. Page 4, line 35: Suggest the first sentence should be revised since it is not only the act of childbirth that contributes to the range of challenges identified – newborn care is also physically and emotionally demanding! Perhaps something along the lines of ‘Following childbirth and during the postpartum period, many women experience...’

16. Introduction and/or Discussion: It might be helpful to reflect on findings from the 2019 qualitative reviews of women’s experiences and preferences during childbirth (<https://journals.plos.org/plosone/article?id=10.1371/journal.pone.0194906>) and the postnatal period (<https://journals.plos.org/plosone/article?id=10.1371/journal.pone.0231415>)

17. Page 5, line 10: Suggest rephrase “economics” – I think “economic” would be better, or “demographic and socio-economic”

18. Page 5, line 23: Suggest “easier” rather than “easy” resolution of conflicts

19. Page 5, lines 23-25: Who is providing “social, financial and emotional support” to whom? If the evidence base describes a positive impact from mutual support, I think it is good to clarify that. If the evidence base reports purely on men’s support of women, good to clarify that too.

20. Page 5, lines 16-35: The authors may wish to consider including feedback loops in the overview of possible mechanisms linking higher relationship quality and HrQoL, e.g. improved women’s psychological HrQoL may contribute to improved relationship quality, in turn contributing to further increases in HrQoL. To consider in the Discussion: the possible existence of a feedback loop may also explain the weaker linear association between the psychological domain of HrQoL and relationship satisfaction.

21. Page 5, line 41: Typo – “Sub-Saharan African”

22. Page 7, lines 32-53: Consider describing the WHOQUOL-BREF domains and items in a separate text box or table

23. Page 11, line 52 (Table 1): Can “Men’s age (in years)” be replaced with “Male partner’s age (in years)?”

24. Page 13, lines 28-35: This could be explained more clearly, and p-values should also be stated. For example, could the first sentence be rephrased as “We also observed no difference in mean scores in the psychological domain of HrQoL between women reporting high compared with very low relationship satisfaction ($p > ??$), and no difference in mean scores in the social domain between women reporting moderate compared with very low relationship satisfaction ($p > ??$).” I also think it might be worth

	editing the paragraph overall to more clearly signpost that there is a linear dose-response relationship within each domain and overall HrQoL even though for several of the domain-specific 'steps' there are no significant differences in HrQoL scores. 25. Page 16, lines 40-42: I suggest not only sociodemographic and economic differences between individual women, but also differences in infrastructure, accessibility of local health-promoting resources, and amenity of local environments – also attributable to differences in country income. 26. Page 16, line 52: I question the authors' description of postnatal depression as a childbirth-related health issue. The causes of postnatal depression are complex and not always (and rarely completely) directly linked to childbirth alone. 27. Page 17, lines 23-30: The study by Chinweuba and colleagues sounds really interesting. The findings presented in this paper seem to illustrate the importance of stressors, rather than only time since childbirth, to HrQoL. Whereas the authors seem to currently be using these findings to illustrate that there is variation in the extent to which HrQoL improves during the postpartum period/extended postpartum period. From the title of the Chinweuba study I gather that they compared HrQoL between women employed in paid work and those not in paid work. Does the HrQoL of women who are not transitioning back to paid work between 12 and 18 weeks postpartum also decline from 12 to 18 weeks, or does it remain stable or perhaps improve? 28. I suggest the Conclusion could be strengthened. A more specific summary of key study findings could be added (e.g. there is an association between women's relationship satisfaction and their HrQoL). More specific implications of study findings could be drawn out – currently it is not clear how the recommendation for increased attention to policies supporting male involvement in joint decision-making and men's support for female partners during the postpartum period has emerged specifically from the results of the study. The authors may also wish to draw out the implications of study findings in terms of developing a more holistic and in-depth understanding of how interventions to influence male partners' engagement in maternal and newborn health can achieve change.
--	---

VERSION 1 – AUTHOR RESPONSE

Reviewer: 1

Dr. Hasagani Tissera, McGill University

1) In the introduction, the authors mention that there is already research examining the link between relationship satisfaction and HrQoL. Therefore, the authors could should make a more compelling case as to why the present research is novel, and what the contributions to the field are.

Our response: In the box containing the study strengths and limitations, we have the following statement:

'This is one of the few studies to explore HrQoL during the extended postpartum period in an African

setting, whereas most studies have documented it during pregnancy or the immediate postpartum period' (page 3, lines 7-9).

Also, we state in the introduction that studies investigating relationship satisfaction in general and in Sub-Saharan Africa and HrQoL are few and with small sample sizes (page 5, lines 14-16 and 20-21). Thus, our study helps to reproduce these results in different contexts and populations and enhance our understanding of the relationship between relationship satisfaction and postpartum women's HrQoL. We have now revised our conclusion to include the following statement: 'Relationship satisfaction is associated with improved HrQoL among postpartum women in Burkina Faso, an important finding for public health practitioners who seek to improve women's wellbeing after childbirth. In another study, we found that an intervention promoting male partner involvement in maternity care in the same population improved relationship satisfaction, alongside uptake of maternity care. It is therefore plausible that policies that support male involvement in maternal health might also lead to increased HrQoL directly or through relationship satisfaction. Further observational and intervention research would be needed to confirm this link and other determinants of women's HrQoL' (page 21, lines 2-10).

2) In terms of the analyses, can the authors provide a justification as to why they summed the relationship satisfaction score and created categories for analyses instead of using the continuous variables?

Our response: There was no particular reason for this in the original study, except that it enabled the authors to use the same analytical approach for all the RCT outcomes.

However, we have now analysed relationship satisfaction as a continuous variable (Table 1 and Table 4). We do recognize that categorization of relationship satisfaction may lead to loss of data sensitivity and thus produce biased estimates of the association between the variables. Indeed, when we analysed relationship satisfaction as a continuous variable (Table 4), we found strong evidence of association between relationship satisfaction and the psychological domain while evidence of such an association was weak when we categorized relationship satisfaction.

We have added the following statement: We analysed relationship satisfaction as a continuous variable (page 10, line 3)

3) The authors employed multiple regression analyses for their main analyses. I wondered whether the different domains of the HrQoL index were highly correlated, which could lead to biased results. Perhaps the authors could include a correlation matrix in which they report the associations between domains, the links between the domains and the other confounders that were included in the analyses.

Our response: We did run the spearman's rank correlation coefficient in our initial analysis. There was evidence of moderate correlation between the physical and psychological domains, physical and environmental domains, psychological and environmental domains, psychological and social domains, and social and environmental domains (page 14, lines 8-13; Table 3, page 14). We have now included these results in Table 3. Since these domains are measuring a similar concept, we expect them to have some correlation.

We did not include the correlation matrix between the domains and the confounders included in the analyses for two reasons: 1) the confounders were categorical and 2) such a matrix table would be too large and look unnecessarily complex without adding any additional meaning to the results

We ran a multicollinearity test in the preliminary analyses to ensure the potential confounders are not highly correlated and did not find the variables to be multicollinear.

4) Relatedly, having relationship satisfaction in this correlation matrix could be beneficial to see how it relates to each of the domains on its own (without controlling for the confounders).

Our response: We have this included in Table 4 in the unadjusted model in which we run a multiple linear regression model with robust standard errors (page 14, lines 16-17 and page 15, lines 1-14; Table 4, page 15)

I also wondered whether the link between relationship satisfaction and the psychological well-being domain was significant in univariate analyses and whether it is the parsing out of the shared variance between this and the other variables in the analyses that weakened this association.

Our response: Thanks to the concerns of the reviewers, when we analysed relationship satisfaction as a continuous variable, we found a very strong evidence of association between the psychological domain and relationship satisfaction at both the univariate and multivariate analyses (page 14, lines 16-17 and page 15, lines 1-14; Table 4, page 15). The weak evidence of association noted in the earlier analysis was most likely due to categorization of relationship adjustment scores which might have led to loss of data sensitivity.

We have added the following statement: We analysed relationship satisfaction as a continuous variable (page 10, line 3)

5) Do the authors have a variable that is indicative of whether the women were primiparous or not? This would be an important confounder to control for. If the authors do not have this variable in the data, they could acknowledge this in the discussion section and speculate whether the present results would generalize to both primiparous and multiparous women.

Our response: We have included this in the new analysis with parity analysed as categorical variable. We have added the following sentence in the methods section: ... women's parity (primiparous – first child for childbirth, secundipara – second child for that childbirth, multipara – two-plus children for that childbirth) (page 10, lines 10-11; Table 1, page 13)

6) Have the authors considered whether the type of marriage (polygamous vs monogamous) could moderate the link between relationship satisfaction and HrQoL?

Our response: Yes, although we did not initially perform an analysis to examine whether type of marriage modifies the relationship between relationship satisfaction and HrQoL, we have done it and there is no evidence of effect modification by type of marriage of the association between relationship satisfaction and HrQoL. We noted the following p-values for the test of effect modification between type of marriage and relationship satisfaction: physical domain =0.272; psychological domain =0.734; social domain =0.764; environmental domain =0.388.

We have not included this in the revised paper as we believe there are many mediators of the association between relationship satisfaction and HrQoL. We could have done it but this was not the scope of this paper.

7) Can the authors discuss and interpret the effect sizes that were found in the present study and compare it to past work? Are these effect sizes comparable to previous work? If not, can the authors speculate as to why?

Our response: For the HrQoL scores, we did provide a comparative study in the discussion section although the study was from a high-income country (Australia) and we have provided an explanation on the same (page 17, lines 9-23).

However, for relationship satisfaction score and HrQoL, we are unable to speculate on the exact effect sizes other than reporting on findings from studies elsewhere. This is because we did not find studies that used the same tools for analysing the relationship between relationship satisfaction and HrQoL that our study used

8) In the abstract and the conclusion section, the authors recommend implementing more policies that are aimed at increasing male involvement during the process of childbirth and child care during the postpartum period. However, in the present study, only women's perception was measured and therefore,

it is not clear whether the findings are driven by actual male involvement. Perhaps, simply perceiving that the partner is involved could be sufficient to produce the results. The authors could discuss this possibility in greater detail in the discussion section, and what it could mean in terms of the type of policies that should be encouraged.

Our response: We recognize that it is difficult to know whether the findings are driven by actual male involvement or not. However, given that in the primary study male involvement did improve women's relationship satisfaction, it is plausible that male involvement interventions, such as the one tested in the RCT, have an effect on quality of life through perhaps a mediating effect of relationship satisfaction. We have now revised our conclusion statement: 'In another study, we found that an intervention promoting male partner involvement in maternity care in the same population improved relationship satisfaction, alongside uptake of maternity care. It is therefore plausible that policies that support male involvement in maternal health might also lead to increased HrQoL directly or through relationship satisfaction. Further observational and intervention research would be needed to confirm this link and other determinants of women's HrQoL' (page 21, lines 2-10).

Reviewer: 2

Ms. Elizabeth Comrie-Thomson, Burnet Institute, Monash University

Comments to the Author:

Our response: We have addressed these in the respective questions raised by the reviewer.

Major revisions

None. The authors are to be congratulated on their work.

Minor essential revisions

1. Page 6, lines 10-38: While I appreciate that the study design is described fully elsewhere, for the analysis in this paper I think it would be appropriate to clarify whether (and on what basis) the authors are confident that there was no contamination between intervention and control arms. And was the analysis an intention-to-treat or per protocol analysis?

Our response: The RCT intervention was provided only to named individuals who received personalised invitations to attend a series of three educational sessions in the health facility, and health workers checked the names of those attending against a list of people invited. While some contamination due to social interactions between individuals in the community cannot be ruled out, these measures will have considerably reduced contamination between the study arms. The analysis was by intention-to-treat.

We have added the following sentence: 'The RCT intervention was provided only to named individuals who received personalised invitations to attend a series of three educational sessions in the health facility, and health workers checked the names of those attending against a list of people invited to minimise contamination between the study arms due to social interactions between individuals in the community' (page 6, lines 15-20)

2. Page 7, lines 6-9: Can the authors please provide more detail on why informed consent was not required for participant data to be used in this analysis and publication? For example, did participants' informed consent for the original study include consent for their de-identified data to be used for secondary analyses?

Our response: The consent forms did not explicitly state that the de-identified data would be used for secondary data analyses (they probably should have!), the analysis for this paper broadly fits with the

overall aim of the study mentioned in the information sheet provided to participants as part of the consent process: “The aim of the study is to understand the role played by husbands or partners of pregnant women in the use of health services after childbirth. We are interested in understanding whether informing men can make them more sensitive to demands of their wives.” The UK and Burkina Faso PIs are authors on this paper (MD, RG, VF), with SC also part of the original investigating team. We have added the following sentence: ‘The dataset has been published, as required by the funder (DFID), on the LSHTM Data Compass repository following de-identification (e.g., all personal names removed, dates changed to month/year format and health centre names anonymised). Researchers can apply to use the data which can be released at the discretion of the study authors. Only the variables needed for the proposed secondary analyses are released. This secondary analysis was done after receiving additional ethical clearance from LSHTM’ (page 7, lines 12-17)

3. Page 7, line 25: Has the French version of WHOQOL-BREF been validated in French-speaking populations? I assume this is the case but it should please be clarified.

Our response: We have addressed this and added the following sentence: Data on HrQoL ... ‘which has been validated among Fench-speaking populations’ (page 8, lines 1-2)

4. Page 8, line 32: It is not clear whether the adapted items were pretested in the population where they were to be used. Could this be specified? And were items revised following pretesting? How were translation and back-translation managed during the pre-testing and revision process?

Our response: We have answered this as follows: All the questions were translated and back-translated from French into Dioula and Moore during collaborative workshops involving the principal investigator, the local field supervisor and the five interviewers. The questionnaires were then pre-tested by the interviewers with acquaintances or women in their neighbourhood. They were asked to report back on the intelligibility and cultural appropriateness of the proposed question formulation. Following their feedback small further modifications were made (page 8, lines 6-12)

5. Page 8, line 41: Please clarify whether “agreement” refers to women’s report of the extent to which she and her male partner agree on the listed topics

Our response: Yes

6. Page 9, line 2: The division of categories is not clear to me. Is it based on quartiles of scores, or total score?

Our response: The total score was categorized using the 25th, 50th and 75th percentiles; so that first category was that below the 25th percentile, second category from 25th percentile to less than 50th percentile, 3rd category was from 50th percentile to less than 75th percentile while 4th category was from 75th percentile and above.

7. Methods: I wonder why relationship satisfaction was not used as a continuous variable – can the authors explain their choice to construct relationship satisfaction as a categorical variable based on externally-derived categories? Because relationship satisfaction is a novel composite variable based on multiple items that have not been locally validated (and I also think the authors have not explored the internal structure of the composite variable, e.g. through item response theory?) it seems to me that measuring relationship satisfaction as a continuous variable might be preferable to dividing it according to externally-derived categories.

Our response: While we initially analysed the exposure variable and potential confounders as categorical variables, we now recognize that categorizing relationship adjustment into quartiles led to attenuation of

the evidence of association between the exposure and the psychological domain. We have now decided to analyse the exposure as a continuous variable.

We have added the following: We analysed relationship satisfaction as a continuous variable (page 10, line 3).

8. Page 9, line 23: I gather that women's postpartum place of residence was a binary variable: residing with her parents/in-laws, or residing with her husband. Either here or in the Introduction, could the authors please explain why it's meaningful to combine residing with her own parents and residing with her in-laws? And can the authors please clarify whether the husband would also reside with the woman while she is at her parents/in-laws – was this data collected?

Our response: The variable was coded separately for residing with parents or residing with in-laws in the original dataset. Given the numbers were small, we thought it was justifiable to combine the two given that residence with parents or in-laws usually means that they are abiding to postpartum separation from their husband i.e., he does not stay in the same house as her. This is relevant for family planning and potentially also for relationship satisfaction given that the woman would be relying on other family members for help and may need her husband's support less.

9. Were partnered women who were not married, or single women, eligible to participate in the RCT, and in the secondary analysis? What happened to participants if a woman's relationship or marital status changed during the study period? Can the authors report any data on number of women excluded from the study due to relationship or marital status? This should also be considered in the interpretation of findings. It is surprising to me that all women in the study were married (according to Table 1).

Our response: Women had to be married or in a co-habiting relationship with a male partner in order to be eligible for the RCT. The way they explained it was "married or living with a man as if married". This included any type of marriage (state-sanctioned which is rare, religious only, traditional) or non-married co-habitation, though most women would refer to themselves as "married" with any kind of co-habitation relationship as this is socially desirable. If women became single by 8 months postpartum, data on relationship satisfaction were not collected and therefore these women are not part of the sample analysed. The numbers excluded for this reason are small: only 8 women excluded from the control group (and the same number from the intervention group).

We have now added the following statement: 'Data for this variable was only collected for women who were still staying with their partners or husbands eight months after childbirth. If women became single by 8 months postpartum, data on relationship satisfaction were not collected and therefore these women are not part of the sample analysed (page 9, lines 12-16)

10. Page 12, lines 32-41: Since Cronbach's alpha is designed to report on internal consistency for a single latent concept, then my understanding is that the alpha for the overall HrQoL is less meaningful than the domain-specific alpha coefficients. This should be clarified in the Results section. I also suggest that internal consistency for the overall HrQoL not be described as "excellent" without also providing a value judgement for the domain-specific alpha coefficients. I also think that the implications of the low alpha coefficients for social and environmental HrQoL domains should be addressed in the Discussion section.

Our response: We have added the following to the discussion section: 'This may be related to the small number of items for the social domain, the unrelatedness of the questions to a concept among this population, or the heterogeneity of the questions. Although the internal consistency of the WHOQOL-BREF tool has been established in the general population, and among postpartum women and found to be satisfactory, it has not been validated among postpartum women in Burkina Faso or Africa. Using a 3-Likert WHOQOL-BREF questionnaire on a large sample of 1,963 Nigerian adolescents, Akpa and Fowobaje showed that a two-factor model of the WHOQOL-BREF had better internal consistency and

reliability than the current universally accepted and used four-factor or domain model. It is thus possible that the differential conceptualisation of the WHOQOL-BREF tool questions may account for the low Cronbach's alpha in the social and environmental domains in our study population' (page 19, lines 1-14)

11. Methods: The authors report on linear associations between relationship satisfaction and HrQoL. Were non-linear relationships tested? Were various associations explored (e.g. dot plot reviewed visually) to confirm whether a linear association was the most appropriate relationship to test?

Our response: Although the data for the physical, psychological and social domains were skewed to the left, we checked for the conditions for multiple linear regression (MLR) and employed robust standard errors to account for the heteroskedasticity that we noted. All the other conditions of MLR were fulfilled. Hence, we did not see the need to check for the non-linear associations or relationships. However, we have added a quadratic term and there is no improvement in the model.

12. Page 15, lines 12-16: Can the authors include one or more citations to support this interpretation of the mechanisms linking shared decision-making, couple communication, and agreement with HrQoL?

Our response: Yes, we have added two references (page 16, line 22).

13. Page 15, lines 31-54 (and continuing on next page): I recommend that the authors consider whether positive feedback loops between higher psychological HrQoL and improved relationship satisfaction (and/or negative feedback loops between lower psychological HrQoL and reduced relationship satisfaction) may provide a plausible explanation for the weaker observed linear association between the psychological domain of HrQoL and relationship satisfaction. The implication of this is that psychological wellbeing, and related concepts of emotional intimacy and emotional support, may be a particularly important aspect of relationship satisfaction among the study population (consistent with the majority of the literature on relationship satisfaction and HrQoL as noted by the authors), rather than a relatively unimportant aspect of relationship satisfaction as currently hypothesised by the authors. I appreciate that the authors note as a study limitation the fact that they cannot rule out reverse causality (page 18, lines 2-4), however I think that this deserves further interrogation in the Discussion section – particularly when seeking to interpret the weaker association seen for the psychological HrQoL domain.

Our response: We acknowledge that reverse causality which could result in a bidirectional link/feedback loop is a possibility and have included this in the discussion section of the revised script. However, it was not possible to investigate with the available data.

In the introduction, we have added the following sentence: 'The relationship between relationship satisfaction and quality of life is complex and maybe reciprocal in which relationship satisfaction may influence or be influenced by quality of life' (page 5, lines 7-9)

In the discussion, we added the following sentence: It is also plausible that a positive feedback loop or reciprocal relationship exists between relationship satisfaction and HrQoL in which relationship satisfaction influences and is influenced by HrQoL, in which, for example, better relationship satisfaction leads to better HrQoL which in turn leads to better relationship satisfaction (page 17, lines 4-8).

It is also true that using relationship satisfaction as a continuous variable gives stronger evidence of association between relationship satisfaction and the psychological domain. We have considered the reviewer's comment and analysed relationship satisfaction as a continuous variable. The results have been presented in Table 4 of the revised manuscript (Page 15)

14. Page 16, line 17 onwards: When comparing study results with findings from other studies, could the authors please describe the timepoint at which other studies' results were captured – i.e. how many months postpartum?

Our response: This has been provided; indeed, it is only the Australian paper that did not have the timepoint at which HrQoL was measured. We have since put the timeline that it was measured at 6 weeks postpartum.

The revised sentence now reads: In a study validating the WHOQOL-BREF tool at 6 weeks postpartum among Australian...' (page 17, line 10)

Discretionary, but recommended, revisions

15. Page 4, line 35: Suggest the first sentence should be revised since it is not only the act of childbirth that contributes to the range of challenges identified – newborn care is also physically and emotionally demanding! Perhaps something along the lines of 'Following childbirth and during the postpartum period, many women experience...'

Our response: This is a very good suggestion that we have incorporated into the article with the following text: 'Following childbirth and during the postpartum period,...'(page 4, line 14)

16. Introduction and/or Discussion: It might be helpful to reflect on findings from the 2019 qualitative reviews of women's experiences and preferences during childbirth (<https://journals.plos.org/plosone/article?id=10.1371/journal.pone.0194906>) and the postnatal period (<https://journals.plos.org/plosone/article?id=10.1371/journal.pone.0231415>)

Our response: We have included in our discussion the following statement/findings from the second paper: 'These findings are also consistent with findings from a systematic review exploring what matters to women in the postnatal period that showed that family relationships as well as overcoming physical and emotional challenges in the postnatal period were important considerations for women' (page 16, lines 9-12)

17. Page 5, line 10: Suggest rephrase "economics" – I think "economic" would be better, or "demographic and socio-economic"

Our response: We have rectified it to "sociodemographic and economic" (page 5, line 4)

18. Page 5, line 23: Suggest "easier" rather than "easy" resolution of conflicts

Our response: Yes, we have rectified it (page 5, line 12)

19. Page 5, lines 23-25: Who is providing "social, financial and emotional support" to whom? If the evidence base describes a positive impact from mutual support, I think it is good to clarify that. If the evidence base reports purely on men's support of women, good to clarify that too.

Our response: The whole paragraph generally focused on support women received from their partners. We have added some words to reflect the fact that the support was from male partners to women. The revised sentence now reads: '... and social, financial and emotional support from the male partner following childbirth and during the postnatal period (page 5, lines 13-14)

20. Page 5, lines 16-35: The authors may wish to consider including feedback loops in the overview of possible mechanisms linking higher relationship quality and HrQoL, e.g. improved women's psychological HrQoL may contribute to improved relationship quality, in turn contributing to further increases in HrQoL.

Our response: In the introduction, we have added the following sentence: 'The relationship between relationship satisfaction and quality of life is complex and maybe reciprocal in which relationship satisfaction may influence or be influenced by quality of life' (page 5, lines 7-9)

In the discussion, we added the following sentence: It is also plausible that a positive feedback loop or reciprocal relationship exists between relationship satisfaction and HrQoL in which relationship satisfaction influences and is influenced by HrQoL, in which, for example, better relationship satisfaction leads to better HrQoL which in turn leads to better relationship satisfaction (page 17, lines 4-8).

To consider in the Discussion: the possible existence of a feedback loop may also explain the weaker linear association between the psychological domain of HrQoL and relationship satisfaction.

Our response: After analysing relationship satisfaction as a continuous variable, we have found stronger evidence of association between relationship satisfaction and the psychological domain. We therefore do not think this is the case. However, we have discussed the possibility of reverse causality or feedback loops as a possible mechanism and cause by which relationship satisfaction influences or is being influenced by HrQoL.

21. Page 5, line 41: Typo – "Sub-Saharan African"

Our response: This has been rectified: Sub-Saharan Africa (page 5, line 20)

22. Page 7, lines 32-53: Consider describing the WHOQOL-BREF domains and items in a separate text box or table

Our response: The WHOQOL-BREF tool is quite lengthy and describing it in a separate text box or table would take up lots of space. We have provided the hyperlink instead (page 8, line 13).

23. Page 11, line 52 (Table 1): Can "Men's age (in years)" be replaced with "Male partner's age (in years)?"

Our response: Done (Table 1, page 13)

24. Page 13, lines 28-35: This could be explained more clearly, and p-values should also be stated. For example, could the first sentence be rephrased as "We also observed no difference in mean scores in the psychological domain of HrQoL between women reporting high compared with very low relationship satisfaction ($p > ??$), and no difference in mean scores in the social domain between women reporting moderate compared with very low relationship satisfaction ($p > ??$)."

I also think it might be worth editing the paragraph overall to more clearly signpost that there is a linear dose-response relationship within each domain and overall HrQoL even though for several of the domain-specific 'steps' there are no significant differences in HrQoL scores.

Our response: We have addressed this in Table 4 of the revised manuscript (page 15)

25. Page 16, lines 40-42: I suggest not only sociodemographic and economic differences between individual women, but also differences in infrastructure, accessibility of local health-promoting resources, and amenity of local environments – also attributable to differences in country income.

Our response: This has been added and the revised sentence reads: The differences are likely due to sociodemographic and economic differences as well as differences in infrastructure, accessibility of local

health-promoting resources, and amenities of the local environments between Australian women (high-income country) and those from our study (low-income country) (page 17, lines 19-23)

26. Page 16, line 52: I question the authors' description of postnatal depression as a childbirth-related health issue. The causes of postnatal depression are complex and not always (and rarely completely) directly linked to childbirth alone.

Our response: We acknowledge the reviewer's insight that the causes of postpartum depression are complex and multifactorial. Here, we did not wish to link it to childbirth but to try to attempt explaining why time may be a factor in HrQoL among postpartum women. Invariably, postpartum depression occurs in the postpartum period and is one of the factors that can affect women's HrQoL. Our use here was to highlight an example of an issue that may affect quality of life in the postpartum period but that which may resolve with time, irrespective of whether it was caused by childbirth or not. On the other hand, a traumatic/difficult childbirth experiences may be either lead to or exacerbate postnatal depression. However, this sentence (and indeed the whole paragraph) has been deleted since it was not necessary when we found a strong evidence of association between relationship satisfaction and the psychological domain after analysing relationship satisfaction as a continuous variable.

27. Page 17, lines 23-30: The study by Chinweuba and colleagues sounds really interesting. The findings presented in this paper seem to illustrate the importance of stressors, rather than only time since childbirth, to HrQoL. Whereas the authors seem to currently be using these findings to illustrate that there is variation in the extent to which HrQoL improves during the postpartum period/extended postpartum period. From the title of the Chinweuba study I gather that they compared HrQoL between women employed in paid work and those not in paid work. Does the HrQoL of women who are not transitioning back to paid work between 12 and 18 weeks postpartum also decline from 12 to 18 weeks, or does it remain stable or perhaps improve?

Our response: Yes, we have since clarified on this point. The paper discusses temporal HrQoL scores between employed and unemployed women at three different timepoints. The findings show that all the women had the highest HrQoL at 12 weeks irrespective of employment status while employed women generally scored lower than their unemployed counterparts at all the three timepoints. The revised sentence now reads: 'In contrast, a study comparing HrQoL among employed and unemployed Nigerian women at 6, 12 and 18 weeks postpartum showed that all women had the highest HrQoL scores at 12 weeks irrespective of employment status while employed women – with only three months' maternity leave – scored lower than the unemployed women at all three timepoints, underscoring the importance of other determinants such as women's employment in addition to temporal influences on postpartum women's HrQoL' (page 18, lines 14-20)

28. I suggest the Conclusion could be strengthened. A more specific summary of key study findings could be added (e.g. there is an association between women's relationship satisfaction and their HrQoL). More specific implications of study findings could be drawn out – currently it is not clear how the recommendation for increased attention to policies supporting male involvement in joint decision-making and men's support for female partners during the postpartum period has emerged specifically from the results of the study. The authors may also wish to draw out the implications of study findings in terms of developing a more holistic and in-depth understanding of how interventions to influence male partners' engagement in maternal and newborn health can achieve change

Our response: We have changed the conclusion to the following statement: "Relationship satisfaction is associated with improved HrQoL among postpartum women in Burkina Faso, an important finding for public health practitioners who seek to improve women's wellbeing after childbirth. In another study, we

found that an intervention promoting male partner involvement in maternity care in the same population improved relationship satisfaction, alongside uptake of maternity care. It is therefore plausible that policies that support male involvement in maternal health might also lead to increased HrQoL directly or through relationship satisfaction. Further observational and intervention research would be needed to confirm this link and other determinants of women's HrQoL". (page 21, lines 2-10)

Reviewer: 1

Competing interests of Reviewer: None declared

Reviewer: 2

Competing interests of Reviewer: I declare that I have no competing interests.

VERSION 2 – REVIEW

REVIEWER	Comrie-Thomson, Elizabeth Burnet Institute
REVIEW RETURNED	16-Aug-2021

GENERAL COMMENTS	I thank the authors for their clear and detailed responses to my initial review. My feedback has been fully addressed in the revised manuscript. I do have one remaining suggestion: since eligible women had to be married or co-habiting with a male partner, then I suggest this be made explicit, and references to marriage in the manuscript be updated to a more inclusive term – perhaps just 'relationship'? I appreciate the detailed explanation provided in the response to my initial review (comment #9), and think it is good to be clear to the reader that unmarried women were not excluded.
--

VERSION 2 – AUTHOR RESPONSE

Reviewer: 2

Ms. Elizabeth Comrie-Thomson, Burnet Institute, Monash University

Our response: On page 10 line 9, Table 1 page 13 and the footnote caption for Table 4, page 15 line 17, we have changed "type of marriage" to "type of relationship".

Reviewer: 2

Competing interests of Reviewer: I declare that I have no competing interests.